# The Prevalence of Fallopian Tube Occlusion in Women with Polycystic Ovary Syndrome Seems Similar to Non-Subfertile Women: A Retrospective Cohort Study

**DOI:** 10.3390/jcm11195610

**Published:** 2022-09-23

**Authors:** Stefan Ghobrial, John Preston Parry, Iris Holzer, Judith Aschauer, Clara Selzer, Andreas Brezina, Samir Helmy-Bader, Johannes Ott

**Affiliations:** 1Clinical Division of Gynecologic Endocrinology and Reproductive Medicine, Medical University of Vienna, 1090 Vienna, Austria; 2Parryscope and Positive Steps Fertility, Madison, WI 39110, USA; 3Department of Obstetrics and Gynecology, University of Mississippi Medical Center, Jackson, MS 39216, USA; 4Privatklinik Goldenes Kreuz, 1090 Vienna, Austria; 5Clinical Division of General Gynecology and Gynecologic Oncology, Medical University of Vienna, 1090 Vienna, Austria

**Keywords:** PCOS, fallopian tube, fertility, tubal occlusion, prevalence

## Abstract

There are limited data on how non-infectious risk factors influence tubal patency in women with subfertility. With hormonal shifts influencing tubal secretions, it has been argued that subfertile women with polycystic ovary syndrome (PCOS) have lower tubal patency. In a retrospective study, 216 women, who underwent diagnostic evaluation for PCOS and infertility, were included. Fallopian tube patency was tested using HSG, HyCoSy, and laparoscopic chromopertubation in 171 (79.2%), 28 (13.0%), and 17 (7.9%), respectively. Bilateral patency was found in 193 women (89.4%), unilateral patency in 13 (6.0%) and bilateral occlusion in 10 (4.6%) patients. Women with PCOS phenotypes C (odds ratio, OR 0.179, 95% CI: 0.039–0.828) and D (OR 0.256, 95% CI: 0.069–0.947) demonstrated lower risks for Fallopian tube occlusion. In conclusion, our data suggest that about 5% of infertile women with PCOS also have bilateral tubal occlusion, which seems similar to the rate in non-subfertile women. With 11% of participants having unilateral or bilateral tubal occlusion, this should reassure women with PCOS that their hormonal challenges do not seem to increase their risk for tubal factor subfertility.

## 1. Introduction

Polycystic ovary syndrome (PCOS) is the most common female endocrinopathy of reproductive age and a frequent cause of female subfertility [1]. In PCOS women with anovulatory infertility, ovarian stimulation with letrozole or clomiphene citrate is first line therapy (after weight loss, if applicable). In 2018, the International PCOS network recommended that for patients with PCOS and infertility due solely to anovulation (and with a normal semen analysis for the partner), the risks, benefits, costs and timing of tubal patency testing should be discussed, whereas tubal patency testing should be considered prior to ovulation induction when there is suspected tubal infertility [2]. However, if anovulatory women do not conceive with ovulation induction, the prevalence of bilateral Fallopian tube occlusion is higher [3]. For this reason, where lightning can arguably be striking the same place twice, it is important to define the yield in exploring unidentified tubal factors in patients with the known risk factor of anovulation.

For the actual prevalence of Fallopian tubal obstruction in women with PCOS, there are few studies. Most prospective randomized studies, which focus on interventions in subfertile PCOS patients, do not report applicable data [4,5,6,7] or use unilateral or bilateral Fallopian tubal occlusion as exclusion criteria without presenting the number of patients excluded for this reason [8,9,10,11,12]. However, the limited available data suggest that bilateral tubal occlusion ranges from 4% in a general subfertile PCOS group [13] to about 8% in PCOS patients needing second line treatment [14].

As previously noted, the risks, benefits, costs and timing of tubal patency testing should be discussed with each individual patient [2]. However, the yield in identifying tubal occlusion is needed to provide an appropriate estimate of benefit. For this reason, we evaluated the prevalence of uni- and bilateral Fallopian tubal occlusion in a population of infertile PCOS women, with exploration of possible predictive factors.

## 2. Materials and Methods

### 2.1. Patient Population

In a retrospective data set, 216 women with ages from 18–40 years underwent diagnostic evaluation for infertility and for PCOS, as defined according to the revised Rotterdam criteria [15]. In detail, hyperandrogenism was defined as a testosterone level > 0.48 ng/mL, which is in accordance with the local normal ranges, and/or the presence of hirsutism [15]. Oligo-/anovulation was diagnosed based on the presence of oligo-/amenorrhea, i.e., a minimum cycle length ≥ 35 days in the last three months. For details about polycystic ovarian morphology (PCOM) see below. Assessment was at the Clinical Division of Gynecologic Endocrinology and Reproductive Medicine of the Medical University of Vienna, Austria, from 1 January 2015 to 31 December 2018. During this interval, all women (including PCOS patients) with infertility were recommended to undergo tubal patency assessment before any infertility-specific treatment. Women who had received any PCOS-specific medication and patients with severe dysmenorrhea (visual analogue scale ≥ 4) were excluded. The study was approved by the ethics committee of the Medical University of Vienna (IRB number 2371/2020, approved on 19 January 2020).

### 2.2. Parameters Analyzed

The AKIM-software (version 7, SAP Software Solutions Austria, Vienna, Austria; SAP-based patient management system at the Medical University of Vienna) was used for data acquisition. The main outcome parameter was Fallopian tubal patency or occlusion diagnosed by either hysterosalpingography (HSG), hysterosalpingo-contrast-sonography (HyCoSy) or laparoscopic chromopertubation, which were performed as published previously [16,17,18].

The following basic patient characteristics were collected: age, body mass index (BMI), duration of infertility, and the type of infertility (primary/secondary). To assess PCOM, an Aloka Prosound 6 ultrasound machine (Wiener Neudorf, Austria; frequency range 3.0–7.5 MHz) was used. PCOM was defined by a follicle number per ovary (FNPO) > 12 and/or an ovarian volume ≥ 10 cm^3^ and/or an ovarian area ≥ 5.5 cm^2^. This is consistent with international recommendations, including use of an ultrasound machine with a frequency range less than 8 MHz [19]. In addition, the following serological data were collected: serum levels of Anti-Mullerian hormone (AMH), luteinizing hormone (LH), follicle stimulating hormone (FSH), total testosterone, dehydroepiandrosterone-sulphate (DHEAS) and sexual hormone binding globulin (SHBG). Blood samples were obtained during the early follicular phase visit (cycle days 2–5). All serum parameters were determined at the Department of Laboratory Medicine, Medical University of Vienna, according to ISO 15189 quality standards. As reported previously [20,21,22], Cobas electrochemiluminescence immunoassays (ECLIA) were performed on Cobas e 602 analysers (Roche, Mannheim, Germany) for the determination of serum FSH, LH, AMH, testosterone, DHEAS, and SHBG. In addition, a Homeostatic Model Assessment of Insulin Resistance (HOMA-IR), calculated as HOMA-IR = insulin (mU/L) × glucose (mg/dL)/405, >2.5 was used for the definition of insulin resistance according to previous studies [23,24].

Moreover, PCOS was classified into the following phenotypes: A, clinical/serological hyperandrogenism + oligo-/anovulation + PCOM; B, clinical/serological hyperandrogenism + oligo-/anovulation; C: clinical/serological hyperandrogenism + PCOM; and D: oligo-/anovulation + PCOM.

### 2.3. Statistical Analysis

Due to the non-Gaussian distribution, which was confirmed using Shapiro–Wilk tests, continuous variables are presented as medians with interquartile ranges (IQR). Categorical data are provided as numbers (frequency). Univariate binary logistic regression models were used to test parameters possibly associated with Fallopian tube occlusion. Only significant parameters would be included in a multivariate binary logistic regression model. Odds ratios (OR) with their corresponding 95% confidence interval (95% CI) are provided for these analyses. The IBM Statistical Package for Social Science software (SPSS 25.0; International Business Machines Corporation, New York, NY, USA) was used for all statistical tests. *p*-values < 0.05 were considered significant.

## 3. Results

Core patient characteristics are provided in Table 1. Phenotypes A, B, C, and D were found in 69 (31.9%), 52 (24.1%), 46 (21.3%), and 49 (22.7%) patients, respectively. Fallopian tubal patency was tested using HSG, HyCoSy, and laparoscopic chromopertubation in 171 (79.2%), 28 (13.0%), and 17 (7.9%) patients, respectively. The 17 patients who underwent laparoscopic chromopertubation had chosen laparoscopic ovarian drilling as their initial PCOS-specific treatment. Using these methods, bilateral patency was found in 193 women (89.4%), unilateral and bilateral occlusion in 13 (6.0%) and 10 (4.6%) patients. The prevalence of tubal occlusion did not differ between the three methods used for its evaluation (Table 2, *p* = 0.220).

Univariate logistic regression analyses were used to evaluate possible factors associated with the presence of any kind of tubal occlusion (either unilateral or bilateral, Table 3). The only parameter which showed a significant association was the PCOS phenotype with phenotypes C and D revealing lower risks for tubal occlusion (with phenotype A as a reference point, phenotype B: OR 0.327, 95% CI: 0.101 to 1.062; phenotype C: OR 0.179, 95% CI: 0.039 to 0.828; and phenotype D: OR 0.256, 95% CI: 0.069 to 0.947; *p* = 0.029). However, ovarian stimulation is recommended for women with PCOS with anovulatory infertility (phenotypes A, B, and D) [2]. Thus, the same model was also calculated including these patients only. Notably, 21/170 (12.4%) of these women revealed uni- or bilateral Fallopian tube occlusion. However, selecting only this group of patients did not alter the findings (Table 3).

When women with PCOS phenotypes B, C, or D (*n* = 147, 68.1%) were compared to those with phenotype A (*n* = 69, 31.9%), the latter revealed significantly higher rates of bilateral tubal occlusion (10.1% versus 2.0%, *p* = 0.013) and any kind of tubal occlusion (either unilateral or bilateral; 20.3% vs. 6.1%, *p* = 0.002). Similar results were found, when only women with oligo-/anovulation were included (Table 4).

## 4. Discussion

In this retrospective cohort study on infertile PCOS women, about 11% of patients had a degree of tubal occlusion. However, only in fewer than half (4.6%) was bilateral tubal occlusion found. It seems noteworthy that all patients underwent a baseline infertility evaluation and that none had undergone previous ovarian stimulation for anovulatory PCOS. Thus, the study did not have detection bias through including women only after they were unable to conceive through ovulation induction. In these women, higher rates of tubal occlusion have been reported [3]. So far, only one study evaluated the prevalence of tubal occlusion in a general subfertile PCOS population and found bilateral tubal occlusion in 4.2% (35/839) [13], which is similar to our findings. Notably, data on non-subfertile women are rare. In one retrospective study which included young non-subfertile women with ovarian cysts, laparoscopic chromopertubation revealed bilateral tubal occlusion also to be 4.2% [25]. Another study focused on infertile women between 18 and 39 years of age with a BMI of ≥29 kg/m^2^ and found bilateral tubal occlusion in 5/574 women (0.9%). In this study, 75% of patients were diagnosed with PCOS [26].

Notably, in many prospective studies bilateral occlusion was used as an exclusion criterion and, thus, PCOS women with unilateral occlusion were included [8,9,10,11,12,13,27,28,29,30,31,32]. However, exact data on its prevalence are rarely provided. In 2014, Legro et al. reported that unilateral occlusion was found in 14.7% (32/217) of those undergoing HSG, but in 64.7% (11/17) when laparoscopy was used, presumably because laparoscopy was reserved for higher risk patients. The overall prevalence of tubal occlusion in this study was 18.4% (43/234) [9], which is high compared to the prevalence of 4.6% in our study population.

Several studies suggested that hyperandrogenism was associated with multiple risks for tubal dysfunction [25,33,34]. In contrast to PCOS-typical serological parameters, PCOS phenotype A was associated with an increased risk for tubal occlusion in our dataset, especially compared to phenotypes B, C and D (Table 3), though the reduced risk in phenotype B was not statistically significant. By comparing women with PCOS phenotypes B, C, or D to those with phenotype A, the latter revealed significantly higher rates of tubal occlusion (Table 4). Especially, the prevalence of bilateral occlusion was relevantly increased (10.1% versus 2.0%, *p* = 0.013). We consider this an important new finding, which may also affect the way patients might be informed in the future.

For study limitations, it is hard to address superimposed non-hormonal risk factors for infection-related tubal occlusion, since many infections are asymptomatic. However, our region is relatively well-funded for public health, lowering the lifetime incidence of pelvic inflammatory disease relative to areas where it is less supported. On a related note, women with severe dysmenorrhea were excluded, making the presence of high stage endometriosis less probable. This could potentially lead to an underestimate for the prevalence of tubal occlusion, since endometriosis has been associated with tubal blockage in PCOS women [35]. Additionally, bilateral tubal spasm or intermittent tubal blockage could not be ruled out for most participants, since patients with proximal tubal blockage diagnosed by either HSG or sonosalpingography can have false positives with open tubes in a subsequent examination [35,36]. However, these women still have a meaningfully lower chance of getting pregnant [36] which makes the findings still relevant.

## 5. Conclusions

In conclusion, our data found 5.6% of infertile women with PCOS to also have bilateral tubal occlusion, which is similar to the rate in non-subfertile women [25]. Coupled with the rate of about 6% for unilateral tubal occlusion in our patient populations, these data can be used for guiding PCOS women on the utility of Fallopian tubal assessment before ovulation induction. The greater PCOS phenotype A seems to be associated with a substantially higher risk for tubal disease. Further studies on a larger scale and in more regions are needed to explore the clinical implications.

## Figures and Tables

**Table 1 jcm-11-05610-t001:** Core patient characteristics and results of hormonal testing.

Age (years) ^1^	30.1 (27.0; 33.6)
BMI (kg/m^2^) ^1^	25.8 (22.1; 29.5)
Primary infertility ^2^	153 (70.8%)
Duration of infertility (months) ^1^	24 (14; 42)
Insulin resistance ^2^	80 (37%)
LH (mIU/mL) ^1^	10.5 (7.4; 15.2)
FSH (mIU/mL) ^1^	5.7 (4.8; 6.8)
LH:FSH ratio ^1^	2.0 (1.3; 2.8)
Testosterone (pg/mL) ^1^	0.42 (0.30; 0.56)
DHEAS (µg/mL) ^1^	2.43 (1.78; 3.41)
SHBG (nmol/L) ^1^	36.8 (24.2; 64.5)
AMH (ng/mL) ^1^	7.3 (5.0; 11.2)

Data are provided as ^1^ median (interquartile ranges) for numerical parameters with 95% confidence intervals or ^2^ percentages for dichotomous categorical parameters. Abbreviations used: BMI, body mass index; LH, luteinizing hormone; FSH, follicle-stimulating hormone; DHEAS, dehydroepiandrosterone sulfate; SHBG, sex hormone-binding globulin; AMH, anti-Mullerian hormone.

**Table 2 jcm-11-05610-t002:** The prevalence of Fallopian tubal occlusion according to the technique used.

	Unilateral Occlusion	Bilateral Occlusion
Hysterosalpingography (*n =* 171)	8 (4.7)	7 (4.1)
Hysterosalpingo-contrast-sonography (*n =* 28)	3 (10.7)	2 (7.1)
Laparoscopic chromopertubation (*n =* 17)	2 (11.8)	1 (5.9)
Total (*n =* 216)	13 (6.0)	10 (4.6)

Data are provided as number (frequency).

**Table 3 jcm-11-05610-t003:** Basic patient characteristics and serological parameters relative to Fallopian tubal occlusion: results of univariate logistic regression analyses.

		All Patients (*n =* 216)		Patients with Oligo-/ Anovulation (*n =* 170)	
		OR (95% CI)	*p*	OR (95% CI)	*p*
Age (years)		1.005 (0.919; 1.100)	0.907	0.983 (0.893; 1.082)	0.728
BMI (kg/m^2^)		0.962 (0.872; 1.060)	0.433	0.973 (0.877; 1.078)	0.598
Duration of infertility		1.008 (0.992; 1.024)	0.315	1.008 (0.991; 1.025)	0.354
Secondary infertility		0.747 (0.300; 1.862)	0.532	0.742 (0.288; 1.909)	0.535
LH (mIU/mL)		1.042 (0.979; 1.108)	0.196	1.047 (0.981; 1.118)	0.169
LH:FSH ratio		1.090 (0.751; 1.582)	0.652	1.086 (0.731; 1.615)	0.682
Testosterone (pg/mL)		2.116 (0.240; 18.635)	0.499	3.174 (0.321; 35.477)	0.311
DHEAS (µg/mL)		1.048 (0.746; 1.471)	0.788	1.062 (0.714; 1.581)	0.765
SHBG (nmol/L)		1.001 (0.987; 1.015)	0.923	0.994 (0.977; 1.012)	0.520
AMH (ng/mL)		1.034 (0.956; 1.118)	0.405	1.045 (0.965; 1.132)	0.277
PCOS phenotype	A	reference		reference	
	B	0.327 (0.101; 1.062)	0.029	0.327 (0.101; 1.062)	0.044
	C	0.179 (0.039; 0.828)		-	
	D	0.256 (0.069; 0.947)		0.256 (0.069; 0.947)	

In the group of all patients, 23/216 women (10.6%) had one or two occluded tubes, whereas this was the case in 21/170 women (12.4%) of the group of oligo-/anovulatory. Abbreviations used: OR, odds ratio; CI, confidence interval; BMI, body mass index; LH, luteinizing hormone; FSH, follicle-stimulating hormone; DHEAS, dehydroepiandrosterone sulfate; SHBG, sex hormone-binding globulin; AMH, anti-Mullerian hormone; PCOS, polycystic ovary syndrome.

**Table 4 jcm-11-05610-t004:** The prevalence of Fallopian tube occlusion in women with polycystic ovary syndrome (PCOS) phenotype A versus women with PCOS phenotypes B, C, or D.

	Phenotype A	Phenotypes B, C, D	*p*
All patients (*n*)	69	147	
Bilateral occlusion (*n*, %)	7 (10.1)	3 (2.0)	0.013
Uni- and bilateral occlusion (*n*, %)	14 (20.3)	9 (6.1)	0.002
Patients with oligo-/anovulation *	69	101	
Bilateral occlusion (*n*, %)	7 (10.1)	1 (1.0)	0.008
Uni- and bilateral occlusion (*n*, %)	14 (20.3)	7 (6.9)	0.010

Data are provided as number (frequency); * for these analyses, there were no patients with phenotype C included.

## Data Availability

The data are available from the corresponding author upon reasonable request.

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
