# Peer review of "The Prevalence of Fallopian Tube Occlusion in Women with Polycystic Ovary Syndrome Seems Similar to Non-Subfertile Women: A Retrospective Cohort Study"

_jcm, 2022, doi:10.3390/jcm11195610_

Round 1

Reviewer 1 Report

The authors present a retrospective cohort study, which is overall well presented and well written. I suggest the following minor changes:

- In the chapter where you describe the parameters analyzed and you classify the PCOS phenotypes, it would be interesting to specify which definition of hyperandrogenism and oligo-anovulation you used to set the diagnosis.

- I suggest to change the title of the article with a more captivating and relevant one. The current title is misleading and suggests a case-control study but you evaluated the prevalence of uni- and bilateral Fallopian tubal occlusion in a cohort of infertile PCOS women and you didn't have infertile non-PCOS patients.

Author Response

The authors present a retrospective cohort study, which is overall well presented and well written. I suggest the following minor changes:

Reply: We thank the reviewer for the appreciative overall assessment of our manuscript.

- In the chapter where you describe the parameters analyzed and you classify the PCOS phenotypes, it would be interesting to specify which definition of hyperandrogenism and oligo-anovulation you used to set the diagnosis.

Reply: We thank the reviewer for this comment. We added the following information to the Methods Section: “In detail, hyperandrogenism was defined as a testosterone level >0.48 ng/mL, which is in accordance with the local normal ranges, and/or the presence of hirsutism [15]. Oligo-/anovulation was diagnosed based on the presence of oligo-/amenorrhea, i.e. a minimum cycle length ≥35 days in the last three months. For details about polycystic ovarian morphology (PCOM) see below.”

- I suggest to change the title of the article with a more captivating and relevant one. The current title is misleading and suggests a case-control study but you evaluated the prevalence of uni- and bilateral Fallopian tubal occlusion in a cohort of infertile PCOS women and you didn't have infertile non-PCOS patients.

Reply: This is a good point. We revised the title as follows: “The prevalence of Fallopian tube occlusion in women with polycystic ovary syndrome seems similar to non-subfertile women: a retrospective cohort study”

Reviewer 2 Report

In this manuscript, the authors assess the prevalence of unilateral and bilateral tubal obstruction in a population of infertile women with PCOS and explore possible predictors. The results of this retrospective study showed that approximately 5.6% of infertile women with PCOS also had bilateral tubal obstruction, and 11% of participants had unilateral or bilateral tubal obstruction, suggesting that hormonal challenge does not appear to increase the risk of developing polycystic ovaries. Syndromic risk of tubal factor infertility in women. Polycystic Ovary Syndrome. This study has certain implications for the effect of PCOS on tubal patency. Further studies on a larger scale and in more regions are recommended to explore the clinical implications.

Author Response

In this manuscript, the authors assess the prevalence of unilateral and bilateral tubal obstruction in a population of infertile women with PCOS and explore possible predictors. The results of this retrospective study showed that approximately 5.6% of infertile women with PCOS also had bilateral tubal obstruction, and 11% of participants had unilateral or bilateral tubal obstruction, suggesting that hormonal challenge does not appear to increase the risk of developing polycystic ovaries. Syndromic risk of tubal factor infertility in women.

Reply: We thank the reviewer for the appreciative overall assessment of our manuscript.

Polycystic Ovary Syndrome. This study has certain implications for the effect of PCOS on tubal patency. Further studies on a larger scale and in more regions are recommended to explore the clinical implications.

Reply: We thank the reviewer for this statement and take the liberty of using his/her wording. We changed the final sentence of the Conclusion Section as follows: “Further studies on a larger scale and in more regions are needed to explore the clinical implications.”